# Radiation and Annealing Effects on GaN MOSFETs Irradiated by 1 MeV Electrons

**Tongshan Lu and Chenghua Wang ***

College of Electronic and Information Engineering, Nanjing University of Aeronautics and Astronautics, 29 Jiangjun Avenue, Nanjing 211106, China; lutongshan@nuaa.edu.cn
* Correspondence: chwang@nuaa.edu.cn

**Abstract:** In this paper, the 650 V N-channel GaN MOSFETs are chosen as the research object to study the radiation and annealing effects under 1 MeV electron irradiation. The output, transfer, and breakdown characteristics are measured before and after electron irradiation. The experimental results show the variation of the I-V curves after irradiation, which is related to the increased conductivity due to the generation of an oxide charge in the GaN MOSFETs. However, the gradual formation of the interface trapped charge offsets the effect of the oxide charge, which decreases the conductivity of the GaN MOSFETs and the drain-source current. The long-term annealing at room temperature degrades the interface trapped charges, leading to the restoration of the I-V characteristics. After room temperature annealing, the breakdown voltage is still higher than the unirradiated level, and this is because the displacement defects caused by electron irradiation cannot be recovered at room temperature.

**Keywords:** GaN MOSFET; electron irradiation; radiation damage; oxide trapped charge; interface charge

## 1. Introduction

As one of the third-generation semiconductor materials, GaN has outstanding intrinsic properties, such as a wide band gap, high critical breakdown electric field, and a high electron mobility and thermal conductivity rate. Currently, GaN-based devices have been widely used in the field of radio frequency (RF), power, and optoelectronic devices [1–3]. GaN MOSFETs avoid the excessive gate leakage current of GaN HEMT and reduce the power consumption [4–6]. It has been paid close attention to by researchers and become the focus of experimental analysis. As a commonly used semiconductor material, GaN-based MOSFET devices play an important role in the fields of automobiles, communications, and measurement. Compared with other bipolar electronic devices, MOSFETs have a faster switching speed, so they are often used in switching devices. Since space electronic systems have a high-power demand, the power systems based on GaN MOSFETs have unique advantages in a space environment. At the same time, the use of space devices has special functions, and they have extremely high requirements for the accuracy of the electronic components used. Therefore, the research on the electrical properties of MOSFET devices has great practical significance.

Space electronic components often operate in extremely harsh environments [7–10]. Microgravity, low temperature, irradiation, plasma, meteor debris, and other space environments will have certain effects on the performance of devices and materials. Among them, the most serious damage to components is mainly irradiation. Energetic particles can interact with the electronic components in the spacecraft, causing ionization effects, displacement effects, and single event effects, causing degradation and even failure of electrical devices [11–13]. As a potential power system application, GaN MOSFETs have received much attention in the aerospace field. Recently, a few studies have investigated

the radiation damage of GaN MOSFETs. The experimental results show that the radiation damage is mainly caused by the ionization and displacement effects induced by the energetic particles in GaN MOSFET [14,15], which will cause the degradation of the electrical performance of the device and even the leakage of electricity. However, further research is needed on the radiation effects and the specific mechanism of irradiation damage in GaN MOSFETs. As a common radiation source, 1 MeV electrons can produce an ionization effect and displacement effect in semiconductor devices, which can be taken as a standard source for radiation evaluation. In addition, high-energy electrons are the main source of particles in the space environment [16,17]. The influence of high-energy electron irradiation on GaN MOSFETs needs to be explored. The analysis of the influence mechanism of the irradiation environment of space on its performance will provide new ideas and directions for the research and development of new MOSFET structure devices in the future, while providing guarantees for the more stable and safe operation of spacecraft and space stations.

In this paper, 650 V N-channel GaN MOSFETs are selected as the research object, and the radiation effects and mechanisms induced by 1 MeV electrons in GaN MOSFETs are investigated. The study results will contribute to the design and manufacture of radiation-hardened GaN MOSFETs in future space applications.

## 2. Experiments

In this study, the 650 V N-channel GaN MOSFETs manufactured in the same batch of processes were selected as the research object. The fabricated GaN MOSFETs were packaged in the TO220 package, and they can operate at a temperature of 150 °C. In order to overcome the device-to-device variation, before irradiation, we checked all the samples to make sure that the difference in the chosen samples was lower than 10%. The datasheet of common parameters for the selected samples is shown in Table 1.

**Table 1.** Datasheet of GaN MOSFET samples.

| Parameter | Parameter Name | Typical Value |
|-----------|----------------|---------------|
| Vgs (TH) | Gate Threshold Voltage | 1.19 V |
| BVdss | Drain-Source Breakdown Voltage | 1345 V |
| Idss | Zero gate Voltage Drain Current, T = 25 °C | 19.5 μA |
| Igss | Gate-Source Leakage | 37.5 μA |
| Rdson | Static Drain-Source on Resistance | 0.215 Ω |

A 1 MeV electron irradiation experiment was performed with a high-voltage electron accelerator at the Technical Physics Institute of the Heilongjiang Academy of Science, China. The fluences were $1 \times 10^{13}$ cm$^{-2}$, $1 \times 10^{14}$ cm$^{-2}$, and $1 \times 10^{15}$ cm$^{-2}$. To ensure that the chip was irradiated during the irradiation experiment, the package of the fabricated GaN MOSFETs was removed to expose the chip completely and prevent electrons from passing through the package. All pins were grounded during irradiation. In space applications, electronic systems are sometimes powered off to save energy. At this time, the components are often grounded at all pins. The study of the radiation effect in this state is helpful for the practical application in space.

Meanwhile, a power characteristic measurement system was exploited to track the electrical characteristics (forward output characteristics, reverse breakdown characteristics, and transfer characteristics) of the GaN MOSFETs during irradiation, including Keithley 2657 A SMU, Keithley 2651 A SMU, and Keithley 2636 B SMU. The performance parameters of the GaN MOSFETs were tested and recorded in the time intervals during irradiation. To prevent the annealing effect at room temperature, the test after each irradiation was completed within five minutes. After electron irradiation, the GaN MOSFETs were subjected to room temperature annealing for 30 min and 48 h. The electrical properties after annealing were studied and analyzed.

In the experiment, the DUT was placed on the test bench for 1 MeV electron irradiation with different fluences. As shown in Figure 1, the sample was connected to the electron-

ical parameter measurement system by cables, which provided the voltage and current conditions required for sample testing. The experimental setup could in situ measure the output and transfer characteristic curves of samples under different electron fluences, which provided strong technical support for subsequent mechanism analysis.

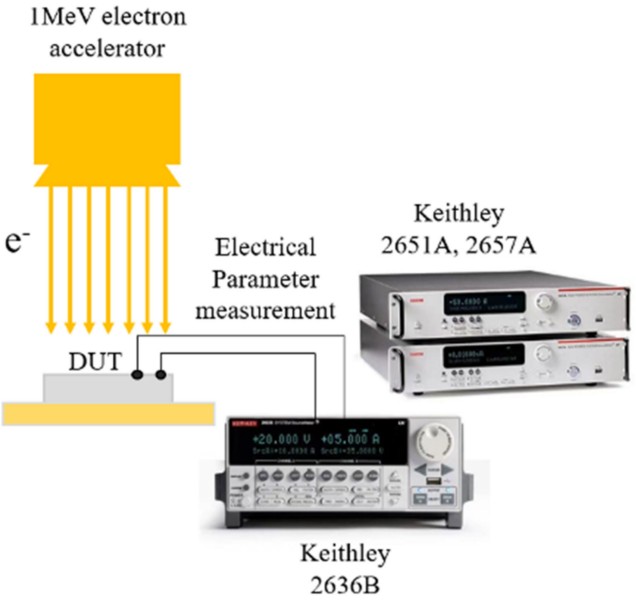

**Figure 1.** Schematic diagram of 1 MeV electron irradiation test.

### 3. Results and Discussion

Figure 2 shows the variation of the $I_{DS}$–$V_{DS}$ curves of the irradiated GaN MOSFET with the various fluences with the gate voltage = 14 V. The curves shown in Figure 2 were extracted from the same irradiated GaN MOSFET sample. From Figure 2, it is shown that the $I_{DS}$–$V_{DS}$ curves retained a linear relationship after irradiation and annealing. It is obvious that the curves tilted up with the increasing electron fluence. However, when the fluence reached $1 \times 10^{15}$ cm$^{-2}$, the curve began to fall back. It can be seen that the drain current decreased sharply after annealing at room temperature for a short period of time. In addition, the $I_{DS}$–$V_{DS}$ curve gradually returned to the unirradiated state with increasing annealing time.

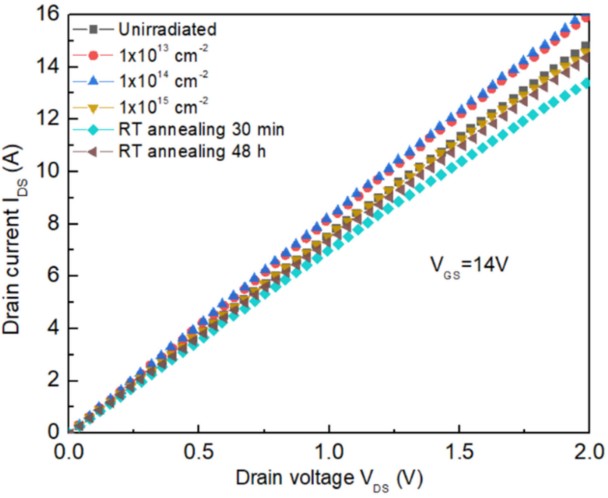

**Figure 2.** The $I_{DS}$–$V_{DS}$ curve of GaN MOSFET before and after electron irradiation.

Since all curves showed a good linear relationship, the $R_{DS(ON)}$ of GaN MOSFETs could be extracted from Figure 2, and the result is shown in Figure 3. The shifts of $R_{DS(ON)}$ illustrated the effect of high-energy electron irradiation on the conductivity of GaN MOSFETs. The $R_{DS(ON)}$ shifted negatively when the electron fluence was less than $1 \times 10^{14}$ cm$^{-2}$. However, the $R_{DS(ON)}$ increased sharply when the fluence was from $1 \times 10^{14}$ cm$^{-2}$ to $1 \times 10^{15}$ cm$^{-2}$. Figure 4 presents the variation of the drain current extracted from Figure 2 when the drain voltage of the devices was 1.5 V. It can be seen from this figure that the drain current increased when the fluence was less than $1 \times 10^{14}$ cm$^{-2}$ and decreased rapidly when the fluence was from $1 \times 10^{14}$ cm$^{-2}$ to $1 \times 10^{15}$ cm$^{-2}$. All the results indicated that electron irradiation improved the conductivity of GaN MOSFETs to some extent but not always.

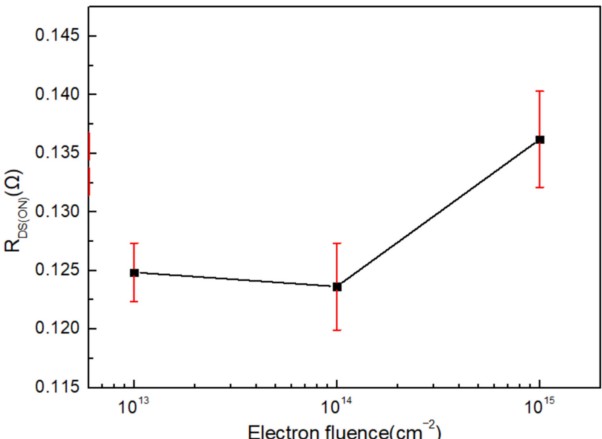

**Figure 3.** The $R_{DS(ON)}$ of GaN MOSFET versus the electron fluence.

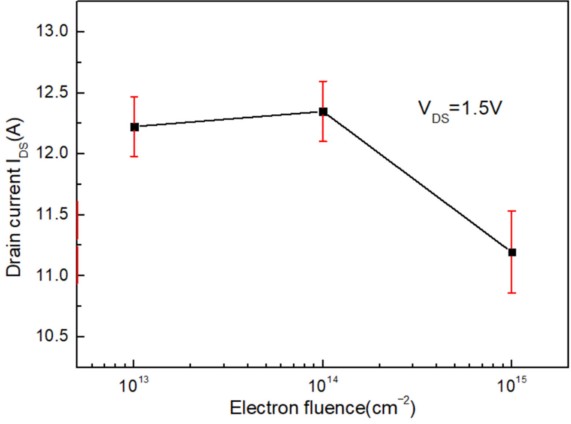

**Figure 4.** Drain current of GaN MOSFET verse the electron fluence as $V_{DS} = 1.5$ V.

This phenomenon should be related to the buildup of positive charges in the oxide, similar to silicon-based MOSFET [18,19]. The increase in the drain current may be due to the large number of oxide charges induced by irradiation, which increased the conductivity of the device. The increase in conductivity led to a decrease in $R_{DS(ON)}$ and an increase in the drain current. However, interface states (negative charges in n-type material) could gradually generate with time, which counteracted the positive charge caused by oxide charges [20]. In this case, the conductivity of the GaN MOSFETs decreased. As for the phenomenon after annealing, this was because the interface trap charges caused by the annealing mad the defects inside the device more obvious, which amplified the negative impact on output characteristics. As the annealing time increased, the interface trap charges gradually disappeared, and their impact on internal defects was reduced.

Figure 5 illustrates the off-state $I_{DS}$–$V_{DS}$ curves measured before and after irradiation. It can be seen that when the source-drain voltage was lower than 400 V, the source-drain current was almost the same at $10^{-7}$ A, while the source-drain current increased sharply when the source-drain voltage was above 400 V. As the electron fluence increased, the avalanche breakdown voltage slightly decreased. The source-drain current of the irradiated MOSFETs in the high voltage part (200 V < V < breakdown voltage) increased with the electron fluence in Figure 5. The experimental results showed that electron irradiation could also improve the drain-source breakdown characteristics of GaN MOSFETs to a certain extent.

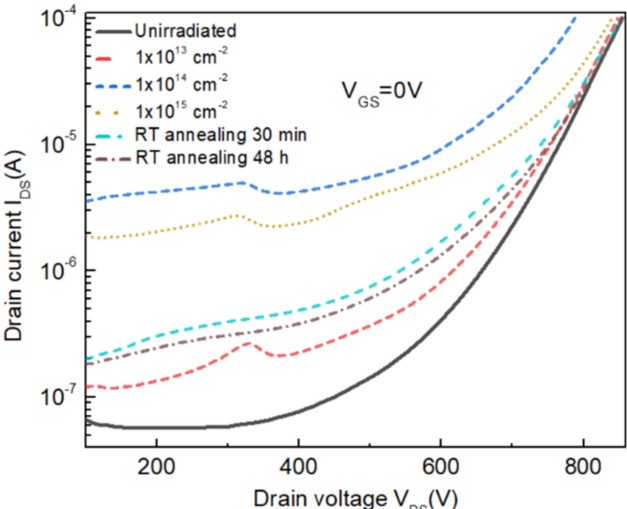

**Figure 5.** $I_{DS}$-$V_{DS}$ curves of GaN MOSFET at off-state before and after irradiation.

When the source-drain voltage was 600 V, the variation of the drain current was as shown in Figure 6. It can be seen that the drain current reached its maximum as the electron fluence reached $1 \times 10^{14}$ cm$^{-2}$. This could be explained by the MOSFET internal leakage due to the generation of atomic displacement and "recombination center". Meanwhile, high-energy electron irradiation also generated oxide charges inside the device, which led to the increase in the drain current. However, when the electron fluence increased to $1 \times 10^{15}$ cm$^{-2}$, the interface trapped charge gradually generated, leading to the decrease in the source-drain current. In addition, Figure 5 also shows that the annealing improved its characteristics as the drain current increased slightly. This may be because the interface trapped charges gradually recovered during the annealing process. Therefore, with the extension of the annealing time, the interface trapped charges gradually disappeared, but the displacement damage was left, making the $I_{DS}$–$V_{DS}$ characteristic at off-state still better than that of the unirradiated ones [18–20].

Figure 7 shows the transfer characteristics before and after electron irradiation, as well as after annealing. It can be seen that the $I_{DS}$-$V_{GS}$ curves shifted negatively, while it shifted positively slightly when the fluence was $1 \times 10^{15}$ cm$^{-2}$. Figure 7 also illustrates that the drain current of GaN MOSFET reached saturation when the gate voltage was above 2.5 V. At the same time, the $I_{DS}$-$V_{GS}$ curve after annealing at room temperature for 30 min had a large positive shift compared to the unirradiated curve. However, the positive shift recovered to the value of the unirradiated curve as the annealing time increased to 48 h.

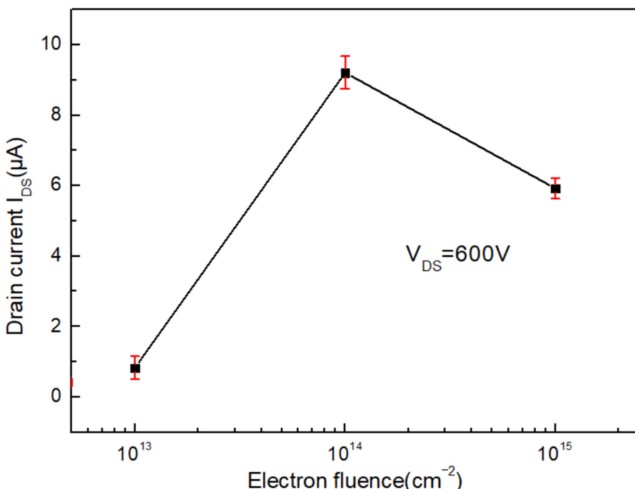

**Figure 6.** Drain current of GaN MOSFET versus the electron fluence with the drain voltage of 600 V.

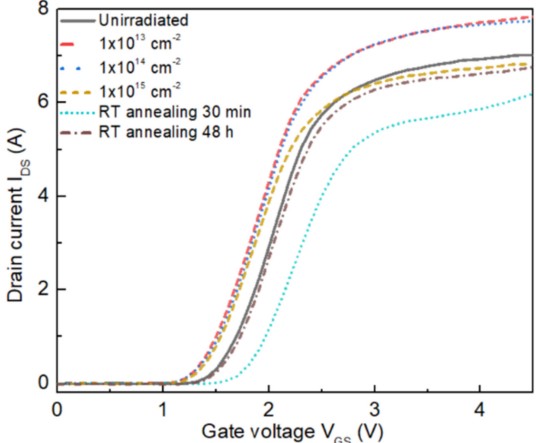

**Figure 7.** Transfer characteristic curves of the GaN MOSFETs before and after irradiation.

Figure 8 presents the variation of the threshold voltage extracted from Figure 7 when the drain current $I_{DS}$ was 0.5 A. As shown in the Figure 8, the threshold voltage of the GaN MOSFETs shifted to the negative side after electron irradiation.

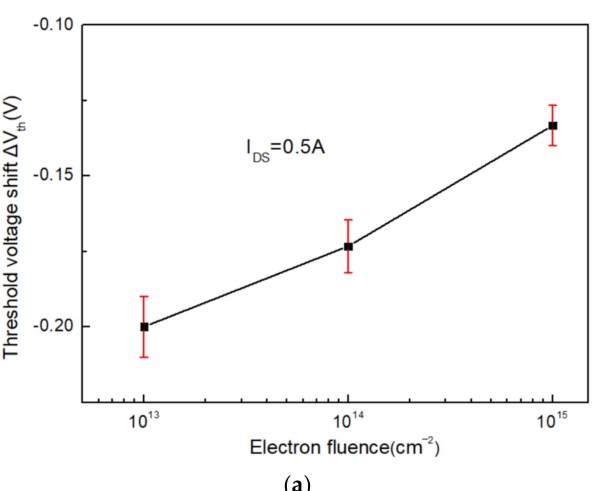

(**a**)

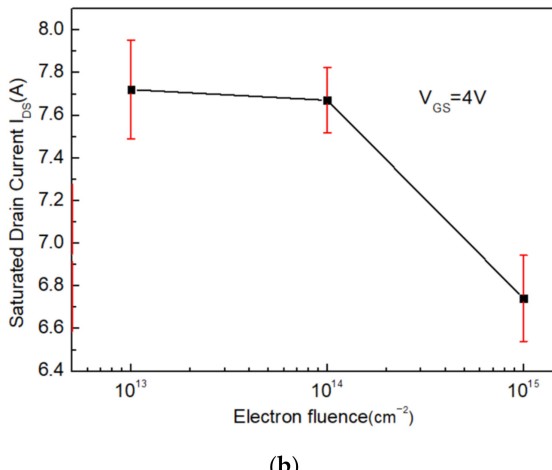

(**b**)

**Figure 8.** (**a**) Threshold voltage shift and (**b**) saturated drain current of the irradiated GaN MOSFETs verse the electron fluence.

The results in Figure 8 show that the variation of the threshold voltage slightly increased after irradiation. However, the threshold voltage remained negative when the electron fluence reached $1 \times 10^{15}$ cm$^{-2}$. Therefore, it is considered that irradiation induced the generation of electron–hole pairs, and the hole in gate oxide had a much slower migration speed than the electron, which caused the oxygen vacancies in the gate oxide to trap holes [20]. Thus, the threshold voltage of the GaN MOSFETs turned to the negative side after irradiation as the oxide trap charges were considered positive. Meanwhile, the generation of oxide charges inside the GaN MOSFETs was much slower than that of the interface trapped charge, which led to the increasing drain current and improved the conductivity of the device after electron irradiation. The gradual disappearance of the interface trapped charge after annealing caused the restoration of the drain current and gate voltage. At the same time, Figure 9 shows the schematic energy band diagram for GaN MOSFET, in which the major physical processes underlying the radiation response are indicated.

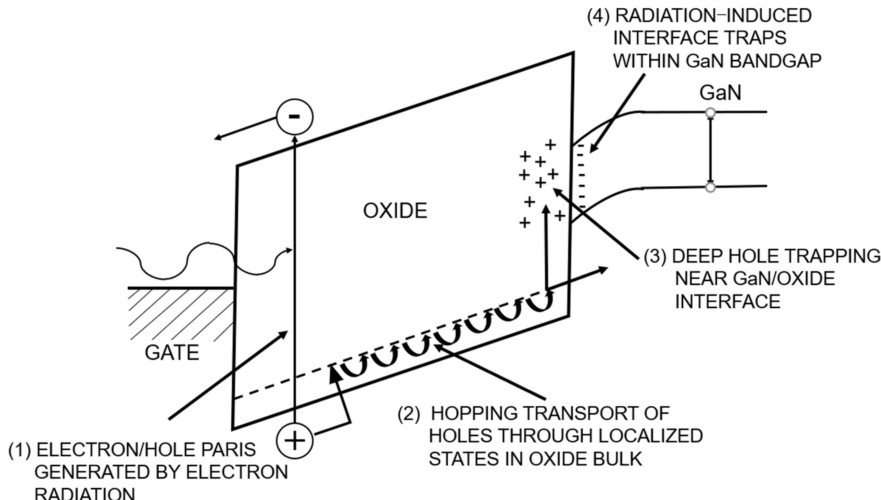

**Figure 9.** Schematic energy band diagram for GaN MOSFET, indicating major physical processes underlying radiation response.

## 4. Conclusions

The experimental results showed that the I-V characteristics (output, transfer, and breakdown) of GaN MOSFETs were affected after high-energy electron irradiation. After irradiation, the annealing treatment at room temperature gradually restored the I-V characteristics of the device to the initial value. This was because the generation of oxide charges was much faster than the interface charges in the GaN MOSFETs during high-energy electron irradiation. Meanwhile, the generation of electron–hole pairs in the gate oxide induced by irradiation caused the oxygen vacancies to trap holes to generate positive charges. The interface charges restored gradually after annealing at room temperature. At the same time, the electrons and holes produced by irradiation were recombined, causing the restoration of the I-V characteristics of GaN MOSFETs. The establishment of the GaN MOSFET and the exploration of specific impact mechanisms still require strong support from subsequent experimental studies.

**Author Contributions:** Writing—original draft preparation, T.L.; writing—review and editing, C.W. All authors have read and agreed to the published version of the manuscript.

**Funding:** This research received no external funding.

**Institutional Review Board Statement:** Not applicable.

**Informed Consent Statement:** Not applicable.

**Data Availability Statement:** Not applicable.

**Conflicts of Interest:** The authors declare no conflict of interest.

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
