# Peer review of "Radiation and Annealing Effects on GaN MOSFETs Irradiated by 1 MeV Electrons"

_electronics, doi:10.3390/electronics11081186_

Round 1

Reviewer 1 Report

Dear authors,

thank you for providing this interesting study on the behavior of irradiated GaN MOSFETs.

I have the following suggestions for improvements, before publishing this paper:

Line 37:

The cited reference [15] does not discuss GaN MOSFETs, but SiC MOSFETs. Please remove this reference or specify correctly.

Fig. 1 and its description in Section 3:

In line 70, you state that “It is obvious that the curves tilt up as electron fluence rises. However, when dose reaches 1×1015 cm-2, the curve begins to fall back.”

For me, it does not become clear if this “tilt up” and “fall back” behaviour lies within the natural scattering of I-V-curves of different GaN MOSFET samples. Please show that the observed changes in the I-V curves after irradiation are statistically relevant effects when compared to unirradiated samples.

Or (maybe I did not fully understand your experimental design): Were the curves shown in Fig. 1 taken from the same GaN MOSFET samples which were irradiated several times with different doses? Then, the change in the I-V curves is obvious. Please make clear how the experiment was performed for a better understanding.

Figs. 2 + 3 + 7:

A log scale on all x-axes would lead to better readability of these figures.

Lines 78 + 79:

Please correct to: RDS(ON) increases sharply when the increasing the dose from 1×1014 cm-2 to 1×1015 cm-2. Same for line 82.

Line 81:

Is this a “leakage” current?

Line 85:

You state that “This phenomenon is mainly related to the buildup of positive charges in the oxide.”

In the context of your paper, this is rather a reasonable assumption, than a proven fact. Because of this; I would weaken the statement in line 85. If there is a matching literature reference for your statement, please refer to it. Then, the statement can be left as it is.

Lines 92-94:

The same here: You formulate this as a statement which is not proven in this paper. Please add a literature reference that confirms your statement or weaken your formulation.

Fig. 5:

Please use a log scale on the x axis and μA units on the y axis.

Lines 104+105 and Fig. 5:

“It can be seen that when the source-drain voltage is lower than 50V”

No, it can not be seen since Fig. 5 starts at 100V.

Or do you want to say 500V?

Line 131:
“The IDS-VGS curves shift sharply as the fluence reaches 1×1014 cm-2”

This is already the case at 1x1013 cm-2, since both curves (1x1013 and 1x1014) are nearly identical. Please describe the figure correctly.

Lines 142-144:

OK. Is there maybe a literature reference which underpins your assumption?

Lines 160-164:

Please make clear that these are reasonable assumptions (unless proven by citing relevant scientific literature).

Author Response

Reviewer 1:

Line 37:

The cited reference [15] does not discuss GaN MOSFETs, but SiC MOSFETs. Please remove this reference or specify correctly.

A: Thanks for the reviewer’s comment. We have deleted the reference [15] in the manuscript, and revised the related statement.

Fig. 1 and its description in Section 3:

In line 70, you state that “It is obvious that the curves tilt up as electron fluence rises. However, when dose reaches 1×1015 cm-2, the curve begins to fall back.”

For me, it does not become clear if this “tilt up” and “fall back” behaviour lies within the natural scattering of I-V-curves of different GaN MOSFET samples. Please show that the observed changes in the I-V curves after irradiation are statistically relevant effects when compared to unirradiated samples.

Or (maybe I did not fully understand your experimental design): Were the curves shown in Fig. 1 taken from the same GaN MOSFET samples which were irradiated several times with different doses? Then, the change in the I-V curves is obvious. Please make clear how the experiment was performed for a better understanding.

A: Thanks for the reviewer’s comment. The reviewer’s thought is right. The curves shown in Fig. 1 is taken from the same GaN MOSFET. We have added the statements about the experiment information of Fig. 1 in the Section 3.

Figs. 2 + 3 + 7:

A log scale on all x-axes would lead to better readability of these figures.

A: We have revised the x-axes of Figs 2, 3, and 7 into a log scale.

Lines 78 + 79:

Please correct to: RDS(ON) increases sharply when the increasing the dose from 1×1014 cm-2 to 1×1015 cm-2. Same for line 82.

A: Thanks for the reviewer’s comment. We have revised the statements in line 78, 79 and 82.

Line 81:

Is this a “leakage” current?

A: Thanks for the reviewer’s comment. We have revised the “leakage current” to “drain current”.

Line 85:

You state that “This phenomenon is mainly related to the buildup of positive charges in the oxide.”

In the context of your paper, this is rather a reasonable assumption, than a proven fact. Because of this; I would weaken the statement in line 85. If there is a matching literature reference for your statement, please refer to it. Then, the statement can be left as it is.

A: Thanks for the reviewer’s comment. The similar mechanism is proven fact in silicon-based MOSFET. We have added some references about Si MOSFET in the manuscript, and weaken the statement in line 85.

Lines 92-94:

The same here: You formulate this as a statement which is not proven in this paper. Please add a literature reference that confirms your statement or weaken your formulation.

A: Thanks for the reviewer’s comment. The similar mechanism is proven fact in silicon-based MOSFET. We have added some references about Si MOSFET in the manuscript, and weaken the statement.

Fig. 5:

Please use a log scale on the x axis and μA units on the y axis.

A: Thanks for the reviewer’s comment. We have revised the Fig. 5.

Lines 104+105 and Fig. 5:

“It can be seen that when the source-drain voltage is lower than 50V”

No, it can not be seen since Fig. 5 starts at 100V.

Or do you want to say 500V?

A: Thanks for the reviewer’s comment. We have changed the “50V” to “400V”.

Line 131:
“The IDS-VGS curves shift sharply as the fluence reaches 1×1014 cm-2”

This is already the case at 1x1013 cm-2, since both curves (1x1013 and 1x1014) are nearly identical. Please describe the figure correctly.

A: Thanks for the reviewer’s comment. We have changed the statements which describe Fig. 6.

Lines 142-144:

OK. Is there maybe a literature reference which underpins your assumption?

A: Thanks for the reviewer’s comment. We have changed the statements and added a reference to support the analysis.

Lines 160-164:

Please make clear that these are reasonable assumptions (unless proven by citing relevant scientific literature).

A: Thanks for the reviewer’s comment. We have changed the statements and added a reference to support the analysis.

Reviewer 2 Report

This is a very well written manuscript, and the effect of radiation exposure is discussed well. I have a few comments before the manuscript publication

  1. The effect of radiation exposure is explained well in the manuscript, but it will be easier for the audience to understand, if the effect can be described with the help of EB diagram of the device. There are various manuscripts in the literature which explained the radiation effect with the help of the EB diagram, for e.g., “Analytical Bit-Error Model of NAND Flash Memories for Dosimetry Application” etc.
  2. In this study, electron irradiation effect is explored, and it is suggested that the study results will contribute to the design and manufacture of radiation harden GaN MOSFETs in future space applications. In the space environment there will be other particles as well, did the author do the comparison with any other radiation source? There are various articles on TID effects on MOSFETs, NAND Flash etc. Is the author planning to extend study by including TID effects or other particles as well?
  3. Please include the error associated with each point for all the figures.
  4. What was the sample size used for the experiment? Was there any device-to-device variation observed? If yes, how did the author overcome that?
  5. On line 57 “All pins were grounded during irradiation” What is the reason to irradiate the chip with all pins grounded? What are the applications of MOSFET in this state?
  6. On line 61, please include datasheet.
  7. On line 71, it is written “when dose reaches 1×1015 cm-2, the curve begins to fall back”, can you please explain the reason behind that?
  8. On line 122, it is written “Therefore, with the extension of the annealing time, the interface trapped charges gradually disappear, but the displacement damage is left, making the IDS–VDS characteristic at off-state still better than that of the unirradiated ones.” Is there any reference for the comment? Please include some citations.
  9. Is there any data after 1*10^15 to support the conclusion of the manuscript?

Author Response

Reviewer2

This is a very well written manuscript, and the effect of radiation exposure is discussed well. I have a few comments before the manuscript publication

  1. The effect of radiation exposure is explained well in the manuscript, but it will be easier for the audience to understand, if the effect can be described with the help of EB diagram of the device. There are various manuscripts in the literature which explained the radiation effect with the help of the EB diagram, for e.g., “Analytical Bit-Error Model of NAND Flash Memories for Dosimetry Application” etc.

A: Thanks for the reviewer’s comment. We have added an EB diagram to explain the radiation effects in GaN MOSFET in the revised manuscript.

  1. In this study, electron irradiation effect is explored, and it is suggested that the study results will contribute to the design and manufacture of radiation harden GaN MOSFETs in future space applications. In the space environment there will be other particles as well, did the author do the comparison with any other radiation source? There are various articles on TID effects on MOSFETs, NAND Flash etc. Is the author planning to extend study by including TID effects or other particles as well?

A: Thanks for the reviewer’s comment. We are planning other radiation source experiments, including Co60 source, protons and so on. The comparison of more radiation source would be studied in the future.  

  1. Please include the error associated with each point for all the figures.

A: Thanks for the reviewer’s comment. We have revised the figures and added the error bar.

  1. What was the sample size used for the experiment? Was there any device-to-device variation observed? If yes, how did the author overcome that?

A: Thanks for the reviewer’s comment. The sample used in this experiment is the 650V N-channel GaN MOSFETs of Foshan GaN-Power company. The fabricated GaN MOSFETs are packaged in the TO220 package. In order to overcome the device-to-device variation, before irradiation, we have checked all the samples to make sure that the difference of the chosen samples is lower than 10%.

  1. On line 57 “All pins were grounded during irradiation” What is the reason to irradiate the chip with all pins grounded? What are the applications of MOSFET in this state?

A: Thanks for the reviewer’s comment. In space applications, electronic systems are sometimes powered off to save energy. At this time, the components are often grounded at all pins. The study of the radiation effect in this state is helpful to the practical application in space.

  1. On line 61, please include datasheet.

A: Thanks for the reviewer’s comment. We have added the datasheet in the revised manuscript.

  1. On line 71, it is written “when dose reaches 1×1015 cm-2, the curve begins to fall back”, can you please explain the reason behind that?

A: Thanks for the reviewer’s comment. This phenomenon should be related to the buildup of positive charges in the oxide, similar to silicon-based MOSFET. The increase of the drain current may be due to the large number of oxide charges induced by irradiation, which increases the conductivity of the device. The increase in conductivity leads to a decrease in RDS(ON) and an increase in the drain current. However, interface states (negative charges in n-type material) could gradually generate with time, which counteracts the positive charge caused by oxide charges. In this case, the conductivity of the GaN MOSFETs decreases. The mechanism has added in the revised manuscript.

  1. On line 122, it is written “Therefore, with the extension of the annealing time, the interface trapped charges gradually disappear, but the displacement damage is left, making the IDS–VDS characteristic at off-state still better than that of the unirradiated ones.” Is there any reference for the comment? Please include some citations.

A: Thanks for the reviewer’s comment. We have added some references in the revised manuscript to support the analysis.

  1. Is there any data after 1*10^15 to support the conclusion of the manuscript?

A: Thanks for the reviewer’s comment. We have revised the related statements, and recheck the conclusion to avoid the misunderstand.

Reviewer 3 Report

The article presents very interesting research results. In my opinion, there is no description and diagrams of how to perform experiments and measurements. There is no description of the test equipment used. Additionally, you should format the figures 1-7 in a uniform manner. Perform a statistical analysis for the measurements, determine the measurement error, etc. 

Author Response

Reviewer3

The article presents very interesting research results. In my opinion, there is no description and diagrams of how to perform experiments and measurements. There is no description of the test equipment used. Additionally, you should format the figures 1-7 in a uniform manner. Perform a statistical analysis for the measurements, determine the measurement error, etc. 

A: Thanks for the reviewer’s comment. We have added the experimental information in Section 2, and revised the figures, as shown in the revised manuscript.
